# Impact of ANCA-Associated Vasculitis on Outcomes of Hospitalizations for Goodpasture’s Syndrome in the United States: Nationwide Inpatient Sample 2003–2014

**DOI:** 10.3390/medicina56030103

**Published:** 2020-03-01

**Authors:** Charat Thongprayoon, Wisit Kaewput, Boonphiphop Boonpheng, Patompong Ungprasert, Tarun Bathini, Narat Srivali, Saraschandra Vallabhajosyula, Jorge L. Castaneda, Divya Monga, Swetha R. Kanduri, Juan Medaura, Wisit Cheungpasitporn

**Affiliations:** 1Division of Nephrology and Hypertension, Department of Medicine, Mayo Clinic, Rochester, MN 55905, USA; charat.thongprayoon@gmail.com; 2Department of Military and Community Medicine, Phramongkutklao College of Medicine, Bangkok 10400, Thailand; wisitnephro@gmail.com; 3Department of Medicine, David Geffen School of Medicine, University of California, Los Angeles, CA 90095, USA; boonpipop.b@gmail.com; 4Clinical Epidemiology Unit, Department of Research and Development, Faculty of Medicine, Siriraj Hospital, Mahidol University, Bangkok 10700, Thailand; p.ungprasert@gmail.com; 5Department of Internal Medicine, University of Arizona, Tucson, AZ 85721, USA; tarunjacobb@gmail.com; 6Department of Internal Medicine, St. Agnes Hospital, Baltimore, MD 21229, USA; nsrivali@gmail.com; 7Department of Cardiovascular Medicine, Mayo Clinic, Rochester, MN 55905, USA; Vallabhajosyula.Saraschandra@mayo.edu; 8Division of Nephrology, Department of Medicine, University of Mississippi Medical Center, Jackson, MS 39216, USA; jcastaneda@umc.edu (J.L.C.); dmonga@umc.edu (D.M.); skanduri@umc.edu (S.R.K.); jmedaura@umc.edu (J.M.)

**Keywords:** Goodpasture syndrome, ANCA, anti-GBM disease, vasculitis, hospitalization, outcomes

## Abstract

*Background and objectives*: Goodpasture’s syndrome (GS) is a rare, life-threatening autoimmune disease. Although the coexistence of anti-neutrophil cytoplasmic antibody (ANCA) with Goodpasture’s syndrome has been recognized, the impacts of ANCA vasculitis on mortality and resource utilization among patients with GS are unclear. *Materials and Methods*: We used the National Inpatient Sample to identify hospitalized patients with a principal diagnosis of GS from 2003 to 2014 in the database. The predictor of interest was the presence of ANCA-associated vasculitis. We tested the differences concerning in-hospital treatment and outcomes between GS patients with and without ANCA-associated vasculitis using logistic regression analysis with adjustment for other clinical characteristics. *Results*: A total of 964 patients were primarily admitted to hospital for GS. Of these, 84 (8.7%) had a concurrent diagnosis of ANCA-associated vasculitis. Hemoptysis was more prevalent in GS patients with ANCA-associated vasculitis. During hospitalization, GS patients with ANCA-associated required non-significantly more mechanical ventilation and non-invasive ventilation support, but non-significantly less renal replacement therapy and plasmapheresis than those with GS alone. There was no significant difference in in-hospital outcomes, including organ failure and mortality, between GS patients with and without ANCA-associated vasculitis. *Conclusions*: Our study demonstrated no significant differences between resource utilization and in-hospital mortality among hospitalized patients with coexistence of ANCA vasculitis and GS, compared to those with GS alone.

## 1. Introduction

Goodpasture’s syndrome (GS) is a rare but potentially life-threatening autoimmune disease [1,2,3,4,5,6]. The syndrome is usually mediated by autoantibodies directed against the non-collagenous domain of the alpha 3 chain of collagen type IV in the basement membranes of the kidneys and lungs, resulting in glomerulonephritis and alveolar hemorrhage [1,3,4,7,8,9,10,11,12,13,14]. Interestingly, GS has been reported concurrently with several other renal diseases, including anti-neutrophil cytoplasm antibody (ANCA)-associated vasculitis, IgA nephropathy, fibrillary glomerulonephritis, and membranous nephropathy [3,7,8,9,10,11]. This process may explain the association of GS with other disorders that disrupt or modify the structure of the basement membranes of the kidneys (such as ANCA vasculitis and lithotripsy) or lungs (such as smoking), and the induction of humoral responses following T cell-mediated glomerular injury [1,3,4].

Studies have demonstrated that the coexistence of ANCA and GS is recognized to occur at a much higher frequency than expected by chance alone, given the rarity of the individual diseases [3,15,16,17]. This phenomenon was first reported within a few years of the first description of ANCA [3,16,17], and it is clear that the autoantibodies are antigenically distinct and that double positivity is not caused by a cross-reactive anomaly [4,18]. It is postulated that the renal involvement in ANCA vasculitis leads to the exposure of antigens from the basement membrane and the formation of anti-glomerular basement membrane (anti-GBM) antibodies [4,18]. The coexistence of ANCAs (mostly myeloperoxidase (MPO-ANCAs)) with anti-GBM antibodies has been increasingly recognized, with a reported prevalence of 20% to 40% [4,13,19,20,21,22,23,24,25,26,27,28]. With this common coexistence, it is therefore recommended that ANCA and anti-GBM should be analyzed in parallel in patients with renal disease, especially those with extrarenal/pulmonary manifestations [3]. Although it has been shown that patients with coexistence of ANCA and GS have severe kidney injury with rapidly progressive glomerulonephritis (RPGN) [4,22,23], the impacts of ANCA vasculitis on mortality and resource utilization among patients with GS are unclear.

Thus, we conducted this study using the 2003–2014 National Inpatient Sample (NIS) database to assess the impacts of coexistence of ANCA vasculitis on inpatient mortality, and resource utilization among patients with GS in the United States.

## 2. Materials and Methods

### 2.1. Data Source

The 2003–2014 National Inpatient Sample (NIS) was used to conduct this cohort study. The NIS is the publically available, inpatient, all-payer database in the United States. This database was developed and maintained by the Healthcare Cost and Utilization Project (HCUP) under the sponsorship of the Agency for Healthcare Research and Quality (AHRQ). The dataset contained more than 7 million hospitalizations annually, which was obtained from a 20% stratified sample of over 4000 non-federal acute care hospitals in more than 40 states of the United States. A survey procedure using discharge weight provided by HCUP-NIS was used to generate national estimates for 95% of hospitalizations nationwide [29]. This dataset included codes for principal and secondary diagnosis as well as codes for procedures performed during the hospitalization. The HCUP-NIS does not capture individual patients. The database captures all information for a given admission. Institutional Review Board approval was not required due to the publicly available nature of this de-identified database (https://hcup-us.ahrq.gov/DUA/dua_508/DUA508version.jsp#hipaa). These data are available to other authors via the HCUP-NIS database with the AHRQ.

### 2.2. Study Population

All patients with a principal diagnosis of GS for the hospitalization based on International Classification of Diseases, Ninth Revision (ICD-9) diagnosis code of 446.21 were included.

### 2.3. Predictor of Interest

The predictor of interest was the presence of anti-neutrophil cytoplasmic antibody (ANCA)-associated vasculitis. ANCA-associated vasculitis was identified using the ICD-9 diagnosis codes of 446.0 (microscopic polyangiitis (MPA)) and 446.4 (granulomatosis polyangiitis (GPA) with and without eosinophilia).

### 2.4. In-Hospital Treatment and Outcome of Interest

Treatments and outcomes during hospitalization were identified based on ICD-9 codes (Appendix A). Treatments of interest included invasive mechanical ventilation, non-invasive ventilation, and renal replacement therapy. Outcomes of interest included respiratory failure, circulatory failure, renal failure, hematologic failure, sepsis, and in-hospital mortality.

### 2.5. Statistical Analysis

Continuous variables were summarized as mean ± standard deviation. Categorical variables were summarized as percentages. The differences within in-hospital treatment and outcomes between GS patients with and without ANCA-associated vasculitis were tested using logistic regression analysis. The odds ratio was adjusted for age, sex, race, smoking, the presence of hemoptysis, and the use of plasmapheresis. A two-tailed *p*-value of less than 0.05 was considered statistically significant. All analyses were performed using SPSS (version 22.0, IBM Corporation, Armonk, NY, USA).

## 3. Results

### 3.1. Clinical Characteristics

A total of 964 patients were primarily admitted to hospital for GS. Of these, 84 (8.7%) had a concurrent diagnosis of ANCA-associated vasculitis, whereas 880 (91.2%) had GS alone without ANCA-associated vasculitis. Among patients with a concurrent diagnosis of ANCA-associated vasculitis, 54 (64%) had granulomatosis polyangiitis and 30 (36%) had microscopic polyangiitis. The mean age of patients was 54 ± 21 years, and 47% were female. Table 1 shows the clinical characteristics based on the presence or absence of ANCA-associated vasculitis in GS patients. GS patients with ANCA-associated vasculitis were older (mean age 57 vs. 54 years; *p* = 0.13), more likely to be male (54% vs. 47%; *p* = 0.23), more likely to have hemoptysis (42% vs. 27%; *p* < 0.01), but less likely to receive plasmapheresis (32% vs. 40%; *p* = 0.18) than GS patients alone.

### 3.2. In-Hospital Treatments

Table 2 shows in-hospital treatment among GS patients. GS patients with ANCA-associated vasculitis required non-significantly more mechanical ventilation (odds ratio (OR) 1.48; 95% confidence interval (CI) 0.87–2.52), and non-invasive ventilation support (OR 1.94; 0.86–4.35) but less renal replacement therapy (OR 0.67; 95% CI 0.42–1.17) than GS alone.

GS patients with GPA required more mechanical ventilation than GS patients alone (OR 1.88; 95% CI 1.00–3.54). In contrast, GS patients with MPA required more non-invasive ventilation (OR 3.34; 95% CI 1.19–9.41) but less renal replacement therapy (OR 0.40; 95% CI 0.18–0.89) than GS patients alone.

### 3.3. Outcomes

The presence of ANCA-associated vasculitis was associated with non-significantly increased risks of respiratory failure (OR 1.42; 95% CI 0.88–2.29), circulatory failure (OR 1.21; 95% CI 0.46–3.17), renal failure (OR 1.47; 95% CI 0.89–2.43), non-significantly decreased risks of hematologic failure (OR 0.68; 95% CI 0.30–1.52), sepsis (OR 0.75; 95% CI 0.26–2.16), and in-hospital mortality (OR 0.71; 95% CI 0.29–1.74) in GS patients, as shown in Table 3. There was no association between ANCA-associated vasculitis and in-hospital mortality in both patients aged <65 or ≥65 years.

The rates of organ failure and in-hospital mortality in GS patients with GPA and in GS patients with MPA were comparable to GS patients alone.

## 4. Discussion

In this study, we demonstrated that hospitalized patients with coexistence of ANCA vasculitis and GS were more likely to have hemoptysis than those with GS alone. Patients with the coexistence of ANCA and GS required non-significantly more mechanical ventilation and non-invasive ventilation support, but non-significantly less renal replacement therapy and plasmapheresis than those with GS alone. There was no significant difference in in-hospital outcomes, including organ failure and mortality between GS patients with and without ANCA-associated vasculitis. There was no significant difference between in-hospital mortality among hospitalized patients with coexistence of ANCA vasculitis with GS and those with GS alone.

Our study found a difference in age distribution among patients with coexistence of ANCA vasculitis with GS compared to those with GS alone. While there was a higher percentage of patients with GS alone aged ≤39 years old and aged 60–69 years old, there were higher percentages of patients with coexistence of ANCA vasculitis and GS aged 50–59 years old and ≥70 years old. This is likely because ANCA vasculitis is most prevalent in patients >50 years old [30], with the peak age between 65 and 74 years old [31], while it is known that GS has a bimodal age distribution in ages 20 to 30 years old and 60 to 70 years old [1,19,22,32,33].

Previous studies have demonstrated the prevalence of ANCA positivity among GS patients of 20% to 40% [4,13,19,20,21,22,23,24,25,26,27,28,34]. A perinuclear fluorescent pattern (P-ANCA) with anti-myeloperoxidase reactivity predominates in GS patients with reported frequencies of 66% to 81% [7,17,35,36,37]. In contrast, circulating anti-GBM antibodies in ANCA-positive patients has been detected in 8% to 14% of patients [17,22,35]. Although there are limited data on types of ANCA among patients with GS in our database, we successfully identified phenotypes of patients with coexistence of ANCA vasculitis (including GPA and MPA) and GS by diagnosis codes. Although previous reports have shown that some patients with coexistence of ANCA vasculitis and GS have features typical for GPA or MPA [4,22,23], the prevalence of coexistence of GPA or MPA phenotypes with GS was unknown. In this study, we demonstrated that 8.7% of patients hospitalized for GS had coexistence of ANCA vasculitis (either GPA or MPA phenotypes).

Patients with the coexistence of ANCA vasculitis with GS usually present with severe kidney and lung disease requiring aggressive immunosuppressive treatment and plasmapheresis [25]. The findings of our study, using a database of hospitalized patients in the United States, also confirmed that patients with coexistence of ANCA vasculitis with GS had a greater rate of hemoptysis than those with GS alone. However, our study demonstrated no significant differences between resource utilization (plasmapheresis, mechanical ventilator, non-invasive ventilator, and renal replacement therapy) among hospitalized patients with coexistence of ANCA vasculitis and GS, compared to those with GS alone.

There has been conflicting data on the prognostic significance of ANCA positivity [7,27,35,36,38,39]. In a large cohort of 221 Chinese patients with GS, the serum level of anti-GBM antibodies and the coexistence of ANCA were independent predictors for mortality [27]. In a recent study of 43 patients with GS that referred to two departments in England over a 20-year period, the coexistence of ANCA positivity was associated with one-year patient mortality [40]. In contrast, Rutgers et al. [7] compared outcomes of 46 MPO-ANCA-positive patients, 10 patients with coexistence of ANCA with the anti-GBM-antibody, and 13 patients with only the anti-GBM-antibody. In this study from the Netherlands, the investigators reported no significant impact of the coexistence of ANCA positivity on one-year mortality [7]. In this study, we utilized the United States’ inpatient hospitalization data from the NIS database and demonstrated no significant difference between in-hospital mortality among hospitalized patients with coexistence of ANCA vasculitis and GS and those with GS alone.

Patients with coexistence of ANCA vasculitis and GS usually present with rapidly progressive glomerulonephritis (RPGN), have severe renal involvement at diagnosis, and are similar to patients with GS alone but more severe than in those with ANCA vasculitis alone [7,19]. While patients with coexistence of ANCA vasculitis and GS may have crescentic lesions that tend to vary in a range of activity versus chronicity, and crescents in GS alone are usually at the same stage of activity, studies have demonstrated that short-term renal survival is poor and comparable among patients with coexistence of ANCA vasculitis and GS and those with GS alone, despite standard immunosuppression regimens [1,10,24,27].

Our study confirmed that hospitalized patients with coexistence of ANCA vasculitis and GS had similar rates of renal failure and renal replacement therapy when compared to those with GS alone. Data on the impact coexistence of ANCA on the long-term renal prognosis of patients with GS were unclear [25,40]. While some data suggest that patients with coexistence of ANCA vasculitis and GS present late and, thus, are dialysis-dependent at diagnosis [7,19,40], a recent study of 37 patients with coexistence of ANCA vasculitis and GS at four centers in Europe demonstrated that GS patients with coexisting ANCA vasculitis had a greater tendency to recover renal function from being dialysis-dependent after treatment, compared to patients with GS alone [25]. Data on outpatient follow-up were limited in the database. Thus, future large multicenter studies with long-term follow-up are needed to evaluate the impact coexistence of ANCA on the long-term renal prognosis of patients with GS.

There are several limitations in this study. Firstly, although the use of the NIS database allows us to evaluate the impacts of ANCA vasculitis on mortality, and resource utilization among patients with GS, possible inaccuracies in ICD-9 CM coding (especially ICD-9 CM coding for GPA and MPA) may confound results. Secondly, given the administrative nature of the dataset, the data on medications, such as immunosuppressants, was limited in this study. Consequently, we could not assess the potential effects of immunosuppressants, such as cyclophosphamide treatment, on hospital outcomes of patients with coexistence of ANCA vasculitis and GS. Thirdly, this is an analysis of an inpatient U.S. database, and this limits the generalizability to the patient population in other countries. Fourthly, kidney biopsy laboratory data (including anti-GBM-antibody titer and types of ANCA), and clinical courses prior to hospitalization are lacking in the database. It has been shown that ANCA may be detected before the onset of GS, suggesting that ANCA-induced glomerular injury could be a trigger for the development of GS [18]. Due to the limitation of the database, future studies are needed to assess if the duration of the ANCAs preceding the development of GS affects the outcomes of GS patients with coexisting ANCA vasculitis.

## 5. Conclusions

In summary, there are no significant differences between resource utilization and in-hospital mortality among hospitalized patients with coexistence of ANCA vasculitis with GS, compared to those with GS alone.

## Figures and Tables

**Table 1 medicina-56-00103-t001:** Clinical characteristics.

Characteristics	Total	Goodpasture’s Syndrome Alone	Goodpasture’s Syndrome and ANCA	*p*-Value
Number	964	880	84	
Age (years)	54 ± 21	54 ± 21	57 ± 19	0.13
≤39	27%	28%	20%	0.01
40–49	9%	10%	6%	
50–59	15%	14%	24%	
60–69	21%	21%	13%	
≥70	28%	28%	37%	
Male	47%	47%	54%	0.23
Caucasian	65%	65%	64%	0.96
Smoking	10%	10%	8%	0.62
Hemoptysis	28%	27%	42%	<0.01
Plasmapheresis	39%	40%	32%	0.18

ANCA = anti-neutrophil cytoplasmic antibody.

**Table 2 medicina-56-00103-t002:** In-hospital treatment among patients with Goodpasture’s syndrome.

Treatment	Goodpasture’s Syndrome Alone	Goodpasture’s Syndrome and ANCA
Mechanical ventilation	18%	25%
Unadjusted OR	1 (ref)	1.50 (0.89–2.53)
Adjusted OR	1 (ref)	1.48 (0.87–2.52)
Non-invasive ventilation	5%	10%
Unadjusted OR	1 (ref)	2.15 (0.98–4.76)
Adjusted OR	1 (ref)	1.94 (0.86–4.35)
Renal replacement therapy	53%	42%
Unadjusted OR	1 (ref)	0.64 (0.41–1.01)
Adjusted OR	1 (ref)	0.67 (0.42–1.07)

Adjusted for age, sex, race, smoking, hemoptysis, and plasmapheresis. OR = odds ratio.

**Table 3 medicina-56-00103-t003:** Outcomes of patients with Goodpasture’s syndrome.

Outcomes	Goodpasture’s Syndrome Alone	Goodpasture’s Syndrome and ANCA
Respiratory failure	29%	38%
Unadjusted OR	1 (ref)	1.54 (0.97–2.45)
Adjusted OR	1 (ref)	1.42 (0.88–2.29)
Circulatory failure	6%	5%
Unadjusted OR	1 (ref)	1.10 (0.42–2.84)
Adjusted OR	1 (ref)	1.21 (0.46–3.17)
Renal failure	61%	70%
Unadjusted OR	1 (ref)	1.50 (0.92–2.44)
Adjusted OR	1 (ref)	1.47 (0.89–2.43)
Hematologic failure	14%	8%
Unadjusted OR	1 (ref)	0.58 (0.26–1.28)
Adjusted OR	1 (ref)	0.68 (0.30–1.52)
Sepsis	7%	5%
Unadjusted OR	1 (ref)	0.71 (0.25–2.00)
Adjusted OR	1 (ref)	0.75 (0.26–2.16)
In-hospital mortality	8%	7%
Unadjusted OR	1 (ref)	0.92 (0.39–2.19)
Adjusted OR	1 (ref)	0.71 (0.29–1.74)

Adjusted for age, sex, race, smoking, hemoptysis and plasmapheresis.

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
