# Peer review of "Impact of ANCA-Associated Vasculitis on Outcomes of Hospitalizations for Goodpasture’s Syndrome in the United States: Nationwide Inpatient Sample 2003–2014"

_medicina, 2020, doi:10.3390/medicina56030103_

Round 1
Reviewer 1 Report
The study of Charat Thongprayoon et al investigates the impact of ANCA-associated vasculitis on outcomes of hospitalization in patients with Goodpasture syndrome. It is a large-scale study and the manuscript is well-written. My comments are:
- In the Methodology section please give the definition of respiratory, circulatory, hematologic, renal failure and sepsis applied in the Results
- Did the authors have any data concerning the immunosuppressive treatment in both groups of patients? Was the treatment different? If no data is available, this should be added in the limitations of the study.
- Although Goodpasture syndrome has been widely associated with MPO-ANCA there are case series, which report PR3-ANCA. Did the authors have information about the type of ANCA in each patient? May that have influenced the outcome?
- Patients with concomitant ANCA vasculitis had a lower (although statistically non-significant) risk for renal replacement therapy. This finding has been also reported by other studies referred by the authors. How do the authors explain this interesting finding?
- Apart from multiple logistic regression, a subgroup regression of the risk of mortality in patients based on the age group of patients may be interesting.
Author Response
Response to reviewer #1
The study of Charat Thongprayoon et al investigates the impact of ANCA-associated vasculitis on outcomes of hospitalization in patients with Goodpasture syndrome. It is a large-scale study and the manuscript is well-written. My comments are:
Response: We thank you for reviewing our manuscript and for your critical evaluation. We really appreciated your input and found your suggestions very helpful.
Comment#1
In the Methodology section please give the definition of respiratory, circulatory, hematologic, renal failure and sepsis applied in the Results
Response: The development of organ failure during hospitalization was identified using ICD-9 diagnosis codes as shown in Table S1
Organ Failure |
Description |
ICD-9CM |
Respiratory |
Acute respiratory failure |
518.81 |
Other pulmonary insufficiency, not elsewhere classified. Includes - acute respiratory distress, acute respiratory insufficiency, adult respiratory distress syndrome NEC |
518.82 |
|
Acute respiratory distress syndrome after shock or trauma |
518.85 |
|
Respiratory distress NOS |
786.09 |
|
Respiratory arrest |
799.1 |
|
Ventilator management |
96.7, 96.70, 96.71, 96.72 |
|
Cardiovascular |
Shock without mention of trauma |
785.5 |
Shock unspecified |
785.50 |
|
Other shock without trauma (includes hypovolemic Shock) |
785.59 |
|
Cardiogenic shock |
785.51 |
|
Septic shock |
785.52 |
|
Hypotension NOS |
458.8, 458.9, 796.3 |
|
Renal |
Acute kidney injury |
584, 584.5, 584.6, 584.7, 584.8, 584.9 |
Hepatic |
Acute hepatic failure or necrosis |
570 |
Hepatic encephalopathy |
572.2 |
|
Hepatitis unspecified |
573.3 |
|
Hepatic infarction |
573.4 |
|
Hematologic |
Defibrination syndrome |
286.6 |
Acquired coagulation factor deficiency |
286.7 |
|
Other coagulation defect |
286.9 |
|
Thrombocytopenia - secondary or unspecified |
287.49, 287.5 |
|
Metabolic |
Acidosis – metabolic or lactic |
276.2 |
Neurologic |
Transient organic psychotic conditions |
293, 293.0, 293.1, 293.8, 293.81, 293.82, 293.83, 293.84, 293.89, 293.9 |
Anoxic brain injury |
348.1 |
|
Acute encephalopathy |
348.3, 348.30, 348.31, 348.39 |
|
Coma |
780.01 |
|
Altered consciousness - unspecified |
780.09 |
|
Electroencephalogram |
89.14 |
|
Sepsis |
Sepsis |
038.0, 038.10, 038.11, 038.19, 038.2, 038.3, 038.4, 038.40, 038.41, 038.42, 038.43, 038.44, 038.49, 038.8, 038.9, 790.7, 117.9, 112.5, 115.04, 115.14, 115.94, 112.81, 112.83, 003.1, 003.21, 036.2, 036.3, 036.0, 036.1, 036.42, 020.2, 022.3, 098.89, 098.84, 098.82, 995.92, 785.52 |
Comment#2
Did the authors have any data concerning the immunosuppressive treatment in both groups of patients? Was the treatment different? If no data is available, this should be added in the limitations of the study.
Response: The reviewer raised very important point regarding immunosuppressive treatment. Data on immunosuppressive treatment were limited in the database. We agree with the reviewer regarding this important point. The following text has been added in the limitations as reviewer’s suggestion.
“Secondly, given the administrative nature of the dataset, the data on medication such as immunosuppression was limited in this study. Consequently, we could not assess the potential effects of immunosuppression, such as cyclophosphamide treatment on hospital outcomes of patients with coexistence of ANCA vasculitis with GS.”
Comment#3
Although Goodpasture syndrome has been widely associated with MPO-ANCA there are case series, which report PR3-ANCA. Did the authors have information about the type of ANCA in each patient? May that have influenced the outcome?
Response: We did not have information of status of MPO- and PR3-ANCA seropositivity but based on ICD-9 diagnosis, we can distinguish patients who had granulomatosis polyangiitis (ICD-9 446.4), and microscopic polyangiitis (ICD-9 446.0). The following statements have been added to the result section to provide the information about the type of ANCA and the impact of different type of ANCA on in-hospital treatments and outcomes as the reviewer’s suggestion.
“Among patients with concurrent diagnosis of ANCA-associated vasculitis, 54 (64%) had granulomatosis polyangiitis and 30 (36%) had microscopic polyangiitis.
GS patients with granulomatosis polyangiitis required more mechanical ventilation than GS patients alone (OR 1.88; 95% CI 1.00-3.54). In contrast, GS patients with microscopic polyangiitis required more non-invasive ventilation (OR 3.34; 95% CI 1.19-9.41) but less renal replacement therapy (OR 0.40; 95% CI 0.18-0.89) than GS patients alone.
The rate of organ failure and in-hospital mortality in GS patients with granulomatosis polyangiitis and in GS patients with microscopic polyangiitis were comparable to GS patients alone.”
Comment#4
Patients with concomitant ANCA vasculitis had a lower (although statistically non-significant) risk for renal replacement therapy. This finding has been also reported by other studies referred by the authors. How do the authors explain this interesting finding?
Response: The reviewer raises very important point. These findings, however, these findings are controversial among published studies likely due to patient characteristic from each study. For example, opposite to the findings from our study in U.S., Alchi B et al (Nephrol Dial Transplant. 2015;30(5):814-21) reported that all of their double-positive patients were dialysis dependent at presentation and none recovered renal function. The numbers are small, but this may suggest that double-positive patients present late. Their worse overall prognosis may be due to a combination of older age and more extensive glomerular injury mediated by both ANCA and anti-GBM antibodies. Alternatively, it may be because the majority of their patients presented late and were dialysis dependent at diagnosis, with only few recovering renal function.
Comment#5
Apart from multiple logistic regression, a subgroup regression of the risk of mortality in patients based on the age group of patients may be interesting.
Response: We agree with the reviewer. Subgroup analysis examining the impact of ANCA-associated vasculitis on mortality in GS patients based on age groups of < 65 and ≥65 years was performed based on reviewer’s suggestion.
Subgroup |
Crude odds ratio, (95% CI) |
P-value |
*Adjusted odds ratio, (95% CI) |
P-value |
Age <65 years old |
0.47 (0.06-3.54) |
0.46 |
0.37 (0.05-2.84) |
0.34 |
Age ≥65 years old |
1.16 (0.43-3.15) |
0.78 |
1.08 (0.38-3.03) |
0.89 |
*Adjusted for gender, race, smoking, hemoptysis, plasmapheresis
The following statements have been added to the result section as suggested.
There was no association between ANCA-associated vasculitis and in-hospital mortality in both patients aged <65 or ≥65 years.
All authors thank the Editors and reviewers for their valuable suggestions. The manuscript has been improved considerably by the suggested revisions!

Reviewer 2 Report
The paper by Thongprayoon et al highlights the occurrence of Goodpasture's syndrome with or without a concurrent ANCA. The paper is of limited interest but the data can contribute to the scientific field. I have some concerns which are only partially assessed by the authors in the discussion.
Major:
- My main problem with this research setup is that data were collected based on ICD registration. If a patient who is diagnosed with Goodpasture, has the ANCA been measured at all times? In other words, was every Goodpasture patient checked for ANCA?
- The authors speak of ANCA vasculitis but is this based on solely a positive ANCA or was there really a proven small-vessel vasculitis? If the last fact is not the case, the authors need to rephrase and just speak of a ANCA seropositivity and no vasculitis. The authors do mention the different diagnoses in their discussion which is odd, it should be described in the results section how many patients had a real classification of GPA, MPA or EGPA.
Minor:
Line 144: the sentence is incorrect
Author Response
Response to reviewer #2
The paper by Thongprayoon et al highlights the occurrence of Goodpasture's syndrome with or without a concurrent ANCA. The paper is of limited interest but the data can contribute to the scientific field. I have some concerns which are only partially assessed by the authors in the discussion.
Response: We thank you for reviewing our manuscript and for your critical evaluation. We really appreciated your input and found your suggestions very helpful.
Comment#1
My main problem with this research setup is that data were collected based on ICD registration. If a patient who is diagnosed with Goodpasture, has the ANCA been measured at all times? In other words, was every Goodpasture patient checked for ANCA?
Response: The reviewer raises an important point. We agree with the reviewer regarding the limitation of the database based on ICD registration. Although testings for anti-GBM and ANCA are both recommended for patients who present with RPGN or pulmonary renal syndrome in clinical practice, we agree with the reviewer regarding the limitation of the study due to lack of laboratory data in the database. The following text has been added in our limitation as reviewer’s suggestion.
“Fourthly, kidney biopsy, laboratory data (including anti-GBM- antibody titer and types of ANCA), and clinical courses prior to hospitalization were lacking in the database.”
Comment#2
The authors speak of ANCA vasculitis but is this based on solely a positive ANCA or was there really a proven small-vessel vasculitis? If the last fact is not the case, the authors need to rephrase and just speak of a ANCA seropositivity and no vasculitis. The authors do mention the different diagnoses in their discussion which is odd, it should be described in the results section how many patients had a real classification of GPA, MPA or EGPA.
Response: We agree with the reviewer’s regarding this important point. We did not have information of status of MPO- and PR3-ANCA seropositivity but based on ICD-9 diagnosis, we can distinguish patients who had granulomatosis polyangiitis (ICD-9 446.4), and microscopic polyangiitis (ICD-9 446.0). The following statements have been added in result section to provide the information about the type of ANCA and the impact of different type of ANCA on in-hospital treatments and outcomes.
Among patients with concurrent diagnosis of ANCA-associated vasculitis, 54 (64%) had granulomatosis polyangiitis and 30 (36%) had microscopic polyangiitis.
Comment#3
Line 144: the sentence is incorrect
Response: We appreciated the reviewer’s thorough comment. We have correct sentence as reviewer’s suggestion.
All authors thank the Editors and reviewers for their valuable suggestions. The manuscript has been improved considerably by the suggested revisions!
